# Notch, SUMOylation, and ESR-Mediated Signalling Are the Main Molecular Pathways Showing Significantly Different Epimutation Scores between Expressing or Not Oestrogen Receptor Breast Cancer in Three Public EWAS Datasets

**DOI:** 10.3390/cancers15164109

**Published:** 2023-08-15

**Authors:** Luigi Corsaro, Davide Gentilini, Luciano Calzari, Vincenzo Sandro Gambino

**Affiliations:** 1Department of Brain and Behavioral Sciences, Università di Pavia, 27100 Pavia, Italy; davide.gentilini@unipv.it (L.C.); vincenzo.gambino@aslto3.piemonte.it (V.S.G.); 2Centro Diagnostico Italiano, Dipartimento di Genetica Molecolare, 20100 Milano, Italy; 3Bioinformatics and Statistical Genomics Unit, IRCCS Istituto Auxologico Italiano, 20145 Milan, Italy; l.calzari@auxologico.it; 4Ospedale di Rivoli, 10098 Rivoli, Italy

**Keywords:** epimutation score, breast cancer, EWAS, stochastic epimutation, methylation, epigenetic association study, epimutation, Notch, SUMO, ESR

## Abstract

**Simple Summary:**

The study compares two groups of breast cancer (BC) identified by estrogen receptor (ER) expression, using an epimutation score calculated from three public methylation datasets, based on the presence of epimutations and on the deviation amplitude of the methylation outlier value. Firstly, for each dataset, a pathway enrichment analysis was performed on the functional gene region with the highest epimutation score; then, the common pathways were found. A higher and significant epimutation score due to hypomethylation in ER-positive BC is present in the promoter region of the genes belonging to the estrogen receptor signaling (ERS) mediated pathway. This is consistent with an active pathway mediated by estrogen function in the group of ER-positive BC. A higher and significant epimutation score due to hypermethylation in ER-positive BC is present in the promoter region of the genes of the SUMOylation and Notch pathways which are associated with BC pathogenesis and play distinct roles in the two BC subgroups. We speculated that the altered methylation profile play a role in regulating pathways with specific functions in the two subgroups of BC.

**Abstract:**

Oestrogen receptor expression in breast cancer (BC) cells is a marker of high cellular differentiation and allows the identification of two BC groups (ER-positive and ER-negative) that, although not completely homogeneous, differ in biological characteristics, clinical behaviour, and therapeutic options. The study, based on three publicly available EWAS (Epigenetic Wide Association Study) datasets, focuses on the comparison between these two groups of breast cancer using an epimutation score. The score is calculated not only based on the presence of the epimutation, but also on the deviation amplitude of the methylation outlier value. For each dataset, we performed a functional analysis based first on the functional gene region of each annotated gene (we aggregated the data per gene region TSS1500, TSS200, first-exon, and body-gene identified by the information from the Illumina Data Sheet), and then, we performed a pathway enrichment analysis through the REACTOME database based on the genes with the highest epimutation score. Thus, we blended our results and found common pathways for all three datasets. We found that a higher and significant epimutation score due to hypermethylation in ER-positive BC is present in the promoter region of the genes belonging to the SUMOylation pathway, the Notch pathway, the IFN-
γ
 signalling pathway, and the deubiquitination protease pathway, while a higher and significant level of epimutation due to hypomethylation in ER-positive BC is present in the promoter region of the genes belonging to the ESR-mediated pathway. The presence of this state of promoter hypomethylation in the ESR-mediated signalling genes is consistent and coherent with an active signalling pathway mediated by oestrogen function in the group of ER-positive BC. The SUMOylation and Notch pathways are associated with BC pathogenesis and have been found to play distinct roles in the two BC subgroups. We speculated that the altered methylation profile may play a role in regulating signalling pathways with specific functions in the two subgroups of ER BC.

## 1. Introduction

Breast cancer (BC) is the most-common tumour in women around the world [1]. Several classification methods have been used to capture the wide heterogeneity of BC: immunohistochemical techniques, molecular features, histological phenotypes, and gene expressions. Immunohistochemically (IHC), BC can be classified based on the expression of the oestrogen receptor (ER), the progesterone receptor (PR), and the receptor tyrosine-protein kinase erbB-2 (HER2) [2]. The immunohistochemical guidelines recommend that BC is considered ER-positive if at least one percent of the nuclei of BC cells are stained and otherwise ER-negative [3]. ER expression is considered a marker of high cellular differentiation, plays an important role in prognosis, and is a predictive marker of the response to endocrine therapy. Although the two BC groups identified in this way (ER-positive/ER-negative) are not completely homogeneous, they differ in their biological characteristics, clinical behaviour, and therapeutic options [4].

Although BC is known to arise from an accumulation of genetic and epigenetic alterations, the molecular pathogenesis of this tumour is still not fully understood. DNA methylation is one of the best-characterised epigenetic alterations involved in carcinogenesis [5]. There are a growing number of reports demonstrating the importance of epigenetic processes in BC pathogenesis and treatment resistance. Several studies have investigated the DNA methylation pattern in BC using genome-wide arrays and the DNA alteration profiling. DNA methylation changes are considered early events in breast cancer progression and are widely accepted as early molecular markers for the diagnosis, prognosis, and prediction of invasive recurrence [6,7,8,9].

In the scientific literature, different kinds of epigenetic cancer analysis can be found. In most reports, the mean methylation level (differential methylation), calculated for each CpG site in human BC, is compared with that in adjacent non-cancerous breast tissue or in normal breast tissue from cancer-free women [10,11]. Other reports have investigated the presence of epigenetic mutations—also defined as “epimutations” or stochastic epigenetic mutations (SEMs)—in breast tissue or in white blood cells from patients with breast cancer [12,13]. However, different definitions of epimutation have been reported in the literature based on the variability or interquartile range of the distributions of the methylation beta values [14,15], and their biological significance is not yet clear. In any case, both types of epigenetic studies (based on differential methylation or epimutations) usually involve a gene-centred analysis, capturing genes that have an altered mean differential methylation or a different epimutation burden. However, detecting the recurrence of rare alterations often requires a large number of samples and presents an even greater challenge in distinguishing between functionally relevant or “driving” alterations and non-oncogenic “passenger” events that may have no functional impact, particularly in tumour types with a high background of genetic or epigenetic alterations [16]. Pathway-focused analysis, as opposed to a gene-focused one, allows the identification of recurrent altered signals or functions in cancer, based on changes found in different genes belonging to the same pathway, but not altered at the same frequency [16].

This study evaluated the presence of epimutations in the two main groups of BCs (ER-positive/ER-negative) identified by the presence of oestrogen receptors as used in clinical practise and defined by current IHC guidelines. We retrieved data from three publicly available datasets and examined the presence of epimutations for each dataset, weighted by their difference from the interquartile range of the distribution of methylation levels in the samples of the entire dataset. We then calculated an epimutation score aggregating the data per gene region (TSS1500, TSS200, first-exon, gene-body) for each gene in each BC sample. Then, we performed a pathway enrichment analysis through the REACTOME database, based on the genes with a significantly higher epimutation score. Finally, we intersected the pathways found in the three datasets. These molecular pathways were characterised by a significantly higher epimutation score in the promoter regions in ER-positive BC samples due to hypomethylation or hypermethylation. The SUMOylation, the Notch, the interferon-
γ
, and the deubiquitination protease pathways were identified by a significantly higher epimutation score due to hypermethylation. The ESR-mediated signalling pathway, on the other hand, was the pathway with a higher epimutation score due to hypomethylation. Interestingly, for all these pathways, there are many studies demonstrating their role in the development of BC, especially for the SUMOylation and Notch pathways, which are directly or indirectly (by affecting many other pathways) involved in the development, progression, relapse, and treatment resistance of BC. Therefore, we speculate that our findings may highlight the importance of the epimutation process in the pathogenesis of different types of breast cancer and that a deeper knowledge of these pathways could likely lead to new therapeutic options to differentially and specifically treat different BC entities. In addition, it cannot be excluded that different mechanisms of molecular alterations (for example, epigenetic versus gene mutation) could involve different genes and signalling pathways. In fact, we observed that the Notch and the SUMOylation pathways are overloaded with epimutations and do not have a high number of genetic mutations, as confirmed by other gene expression studies [17,18]. Finally, since the epimutations affected a few specific signalling pathways, it cannot be excluded that an apparently stochastic process such as the epigenetic one could be at least partially under deterministic control, as already suggested by other authors [19].

## 2. Materials and Methods

### 2.1. Selection, Downloading, and Preprocessing of the Datasets

For our study, we selected publicly available datasets with methylation profiles of breast cancer tissue in patients of Caucasian ethnicity. The presence of immunohistochemical assessment of oestrogen receptor status on breast cancer cells was a preferred indicator for dataset selection. Other clinical characteristics included patient age and tumour stage, which are generally considered associated with cancer cell methylation status [5,14]. We identified three datasets: TCGA-BRCA and the TGCA-27k from the TCGA data portal [20], with raw idat files, and the GSE69914 dataset [21], from the GEO data portal, with a matrix of beta values already subjected to preliminary quality control.

The TCGA-BRCA and GSE69914 datasets were created using Illumina methylation technology with 450k probes, while the dataset TCGA-BRCA-27k was created using Illumina methylation technology with 27k probes. All three datasets contain information on immunohistochemically identified oestrogen receptor status according to the latest clinical guidelines, while age and staging are only available for the two TCGA datasets.

The characteristics of the three datasets are summarised in Table 1.

The datasets were downloaded using the R GenomicDataCommons 1.24.2 [22] package for the TCGA platform and the R GEOquery 2.68.0 [23] package for the GEO Accession Omnibus [24] platform. All data processing was performed using the R 4.2.2 ecosystem on a server with 32 cores, 128 Gb of RAM, and a 4 Tb hard disk. Data from the TCGA-BRCA and TCGA-BRCA-27K datasets were imported using the ChAMP package 2.30.0 [25], removing probes with missed values, or a detection *p*-value greater than 1%, or with a bead count of less than three in 5% or more of the samples. A sample was removed if more than 10% of the probes were lost due to quality issues. Probes related to SNPs and multi-hit probes were removed. The datasets were also normalised using the SWAN method. Probes on sexual chromosomes remained intact. The GSE69914 dataset is a matrix of beta values that had already undergone preprocessing quality control as indicated in the poster information on the GEO website. Table 1 shows the characteristics and the number of probes of each dataset after quality preprocessing.

After the preprocessing steps, the resulting matrix expresses the beta- value coefficients of methylation. The beta-value method has a direct biological interpretation—it roughly corresponds to the percentage of a site that is methylated. However, from an analytical and statistical perspective, the beta-value method has strong heteroskedasticity outside the mean methylation range, which is a major problem in the application of many statistical models. In comparison, the M-value method, which is roughly equivalent to a logarithmic transformation of the beta-value, is statistically more valid in differential and other statistical analyses because it is approximately homoscedastic and the difference of the M-value can be interpreted as the fold-change [26]. Therefore, we applied a transformation of the beta-value to the M-value to our data and performed our next analysis with the M-values. The M-values were obtained directly via the ChAMP package after a preprocessing check for the TCGA data portal datasets, while we transformed the beta-values of the GEO dataset into M-values via the beta2m function of the lumi R package 2.52.0 [27].

### 2.2. DNA Methylation Analysis

Since DNA methylation is a process known to correlate with age, the biological age for the GEO dataset GEO69914 was derived using the GP-age package, even though this process is only validated for blood and not for all tissue types. We then calculated the epithelial component for each sample using EpiDISH 2.16.0 [28], an R package for deriving the proportions of a priori known cell types in a sample representing a mixture of such cell types. This package can be used for DNAm data from whole blood, general epithelial tissue, and breast tissue. Finally, PCA analysis of the methylation data assessed the correlation of the methylation M-values with age, epithelial components of breast tissue, and tumour stage.

### 2.3. Epimutation Detection

To identify an epimutation, the method described by Gentilini et al. [5] was used. A stochastic epigenetic mutation or “epimutation” (SEM), at a given CpG site, was defined as an extreme outlier of the DNA methylation value distribution across individuals [5]. At the beginning, each dataset was considered as an independent experiment. For each dataset, the distribution of the M-values of each CpG probe in the dataset population was calculated. Then, we obtained the inter-range quantile (IRQ) of the distribution of the M-values of each CpG probe. Then, we defined as epimutated the M-value of a probe in a sample if this M-value was outside the interval defined in the following equations for the lower limit:
(1)
$$L_{min}=Q_{1}-\left(3\times IQR\right)$$

and the following equation for the upper limit:
(2)
$$L_{max}=Q_{3}+\left(3\times IQR\right)$$

where 
$Q_{1}$
 and 
$Q_{3}$
 are, respectively, the first and the third quartile and *IQR* is the interquartile range.

Using the package semseeker [29] with the semseeker function, we calculated the absolute value of the difference (which we called delta, 
δ
) between the M-value of each probe minus the corresponding limit of the interval used to define an epimutation for all probes: 
$|M_{Value}-L_{max}|$
 if it was a hypermethylated probe or 
$|M_{Value}-L_{min}|$
 if it was a hypomethylated probe. Figure 1 shows in a schematic way the definition of epimutation and of delta. The probes with an M-Value within the defined range between 
$L_{max}$
 and 
$L_{min}$
 were set to zero. In turn, we calculated the distribution of the deltas for both hypermethylated probes and hypomethylated probes. We then applied the quartile ranking of absolute delta values to the whole genome and assigned a score from 1 to 4 to each quartile. In this way, we not only determined the presence of a single epimutation, but also ranked the epimutation weight in the whole genome; the higher the deviant value was, the higher was the rank of the epimutation. The quartile ranking was applied to eliminate technical bias in methylation measurement.

Then, all the probes were annotated using the official Illumina manifest file [30], using the hg19 genome version as the reference, in order to quantify the total epimutation weight in each gene region (TSS1500, TSS200, first-exon, body-gene).

Finally, the ranks of the probes belonging to the same gene region (TSS1500, TSS200, first-exon, body-gene) were summed for each annotated gene. At the end, we obtained two synthetic values for each gene region for each sample: one for the hypermethylated probes and one for the hypomethylated probes. We called these two values weighted stochastic epimutation scores (WSEMSs).

Finally, for each gene region, a quantile regression model (at the median) was applied between the WSEMSs obtained for both the hypomethylated and hypermethylated gene regions and oestrogen receptor status using age, epithelial proportion, and clinical stage (available for both TCGA datasets) as covariates.

(3)
$$ER_{status}∼Epimutation_{Burden}+Age+Stage+Epithelial_{component}$$


The quantile regression model was calculated using the R package lqmm [31]. All results were corrected for multiple testing using [32] the Benjamini and Hochberg method.

### 2.4. Identification of Genes and Pathways

Quantile regression results were filtered to identify gene regions with statistically significant epimutation burden (hypomethylated or hypermethylated), and pathway analysis was performed using the pathfindR R package [33]. As a result, pathfindR generates a table with the ID and the name of the pathway resulting from the enriched analysis with the significant gene regions, the lowest adjusted *p*-value of the given term over all iterations (
$lowest_{p}$
), the highest adjusted *p*-value of the given term over all iterations (
$highest_{p}$
), and the number of occurrences of this gene in the pathway over all iterations. Finally, the pathways identified in each dataset were intersected to find the common pathways among all datasets (Section 3.5).

### 2.5. Workflow

For each dataset, the adopted workflow is shown in Figure 2.

## 3. Results

### 3.1. Clinical and Biological Characteristics of BC Patients and Tumour Samples

We began by downloading three publicly available datasets. Table 2 summarises the clinical and biological characteristics of the patients in the three datasets.

Anamnestic personal age is not available in the GSE69914 dataset; therefore, the biological age was derived from the same methylation data using an algorithm based on blood methylation data, although only cancer tissue data were available. The mean age was 57.65 ± 12.74 years in the TCGA-BRCA dataset, 49.93 ± 5.62 years in the GSE69914 dataset, and 59.19 ± 13.11 years in the TCGA-BRCA-27k dataset. The clinical stage was not included in the GSE 69914 dataset. The distribution of the clinical stage of BC in the other two datasets for which it was available was similar and consistent with reports in the literature, with the most part of the cases occurring in the first two stages of BC [1]. As reported in the literature, the ratio of ER-positive to ER-negative BC was also about 20–80% as in Table 3 [34].

The presence of HER2 expression was evaluated in the BC samples of the 450k TCGA dataset (for which these data were available). The most part of BC was HER2-negative, both in ER-positive BC (57.3%) and in ER-negative BC (58%). HER2-positive BC corresponded to 12.7% in ER-positive BC and to 9.5% in ER-negative BC, respectively. The number of HER2+ BC was only a minimal part of the total BCs, and the percentages of HER2+ and HER2− were similar in both groups of ER-positive and ER-negative BCs. Moreover, we executed a Pearson’s Chi-squared test, whose *p*-value was 0.527, confirming a non-statistical significance dependence of the two variables. Therefore, we considered the presence of HER2 non-determinant of the difference between ER-positive and ER-negative, and we did not use HER2 as a covariate in our analysis, as in other similar studies comparing characteristics between ER-positive and ER-negative BC.

### 3.2. Analysis of Methylation Profiles of BC Tissues

After preprocessing analysis, the correlation was evaluated between the methylation profile of BC samples and the following three factors: patient age, clinical stage, and epithelial components of BC samples, which are known variables affecting the methylation profile of BC tissues [5,6]. Therefore, we performed PCA analysis and confirmed that these variables were correlated with the methylation data for the TCGA datasets (Figure 3 and Figure 4). For the third dataset (GSE69914), the PCA correlation of age and methylation profile is shown only for completeness, since age was calculated on the methylation data (Figure 5). Finally, we included the epithelial component, patient age, and clinical stage as covariates in our final regression model (as described in Section 2).

### 3.3. Epigenetic Mutation Analysis and Definition of “Epimutation Score”

In this study, we investigated the association of the epimutation score in two main groups of breast cancer identified immunohistochemically by oestrogen receptor expression according to the latest clinical guidelines. We began by analysing methylation data from one dataset at a time. We determined an “epimutation score” for each gene region as explained in the Materials and Methods Section 2. We then applied a quantile regression model to the median for each gene region for each type of epimutation that occurred (by hypermethylation and hypomethylation). In this way, we obtained the beta regression coefficients explaining how much the epimutation burden differed between the two groups. We plotted the beta regression coefficients for the expression of ER and their corresponding *p*-values in the volcano plots in Figure 6 and Figure 7.

We interpreted the positive beta coefficients (on the right of the vertical axis) as a measure of the higher total epimutation burden in the gene region for ER-positive BC compared with ER-negative BC and the negative beta coefficients (on the left of the vertical axis) as a measure of the higher burden of epimutated probes in the gene region present in ER-negative BC compared with ER-positive BC; this was applied both to hypermethylated and to hypomethylated analyses.

### 3.4. Identification of the Most-Epimutated Genes

Consequently, we filtered out the genes with statistically significant regression beta coefficients in the gene regions studied. The ridge plots in Figure 8 and Figure 9 give an idea of a large number of genes with a statistically significant presence of epimutations, both hypomethylated and hypermethylated.

In this way, we obtained for each dataset a list of genes with different epimutation loads in the two BC groups (one list due to hypomethylation and one due to hypermethylation). Using these lists, we performed pathway enrichment analysis in the REACTOME database, the results of which are shown in Table 4, Table 5 and Table 6 for pathways impacted by genes with Hypormethylation and Table 7, Table 8 and Table 9 for pathways impacted by genes with Hypermethylation. All the pathways found after enrichment for all the dataset are available in Appendix A.

All these steps are summarised in Figure 10 and Figure 11, which are Venn diagrams of the multiple crossing steps.

### 3.5. Identification of Common Pathways

The pathways identified by the enrichment analysis of each dataset were crossed as summarised in the Venn diagrams in Figure 10 and Figure 11. We found common pathways for the three datasets characterised by a higher burden of epimutations in the TSS1500 gene region than for the hypomethylation and hypermethylation probes. The following tables show the pathways shared by all three datasets analysed.

In the first Table 10 are the pathways corresponding to the pathway retrieved from the TSS1500 gene region with epimutated probes that showed significantly higher epimutation levels in ER-positive BC versus ER-negative due to hypermethylation. We can note that most of them belong to two main pathways, the Notch pathway and the SUMOylation pathway, which are associated with breast cancer, as explained in the next section. Two other pathways were the UCH proteinase pathway and the Ub-specific processing Ub-specific processing protease pathway, both of which are related to the regulation of the ubiquitination process, which is another post-translational protein process like SUMOylation. The last pathway was the regulation of signal transduction processes, which have been found to affect BC development in different ways depending on the expression of ER. We discuss the role of these signalling pathways in the development of Lyme disease in more detail. We filtered the pathways to obtain only those present in 90% of the iterations performed by pathfindR.

The second Table 11 lists the pathways corresponding to those retrieved from the TSS1500 gene region with epimutated probes that showed significantly higher epimutation levels in ER-positive BC versus ER-negative BC due to hypomethylation. In this case, the common pathway was ESR-mediated signalling, a known pathway associated with the effects [35,36].

## 4. Discussion

Breast cancer (BC) is the most-common tumour in women. It is a multifactorial disease with a high grade of heterogeneity often contributing to making breast cancer difficult to treat. Different methods of classification, such as the immunohistochemical technique, molecular characteristics, and gene expression, have been used to frame this high heterogeneity in order to foresee the prognosis and to choose the best treatment options [37,38]. Immunohistochemically, BC can be classified by the expression of oestrogen receptors (ERs), progesterone receptors (PRs), and receptor tyrosine-protein kinase erbB-2 (HER2) [2,3,4]. The clinical guidelines for immunohistochemical (IHC) quantitation of steroid receptors in BC recommend that ER and PR assays be considered positive if at least one percent of nuclei are stained [3]. Although the two groups of BC identified in this way (ER-positive/ER-negative) are not completely homogeneous, the two BC groups can be differentiated by biological characteristics and clinical behaviour [39]. It is noteworthy that the tumour ER expression is considered an element of high cellular differentiation and has a very important role in prognosis and therapy [4]. In fact, breast cancer prognosis progressively worsens in ER-negative subtypes due to their high aggressiveness, hormonal therapy insensitivity, and chemoresistance, and a subset of patients will progress to relapse after CT remission, which subsequently leads to metastasis. Furthermore, in patients with ER-positive BC, the relapses have molecular characteristics similar to those of ER-negative BC [39,40]. The underlying mechanisms of BC heterogeneity features and mechanisms that drive therapy resistance (both hormonal and chemotherapeutic) are conundrums that have still to be completely solved, and efforts have to be made in order to better understand the biology of BC and stratify patients to effective treatments [39].

In our study, we tried to characterise these two groups of breast cancers (ER-positive and -negative) by applying an epigenetic score based on the identification of different epigenetic outliers (defined as epimutations). An epimutation, at a given CpG site, could be defined as an extreme outlier of the DNA methylation value distribution across individuals [15]. Previous studies evaluated the presence of epigenetic outliers in BC, but they compared BC tumour samples vs. normal breast tissue or blood samples from BC patients vs. control women without BC [15,16,41]. Teschendorff AE et al. [16] demonstrated that DNA methylation outliers in pre-neoplastic lesions define epigenetic field defects, marking cells that become enriched in invasive breast cancer and cervix cancer and that may, therefore, contribute casually to cancer progression. In another study, the same group highlighted that the identification of outlier methylation profiles allows more-reliable identification of risk-associated CpGs than statistics based on differences in mean methylation levels [41].

### 4.1. Cancer Cells, Epigenetic Mechanisms, and DNA Methylation

Cancer cells acquire the ability to divide and grow uncontrollably [17]. Though it is well established that this could be due to both genomic and epigenetic alterations, the process through which cells acquire this characteristic is not completely understood [42]. Several studies have demonstrated the importance of epigenetic alterations in multiple aspects of cancer biology (tumour pathogenesis and immunomodulation), cancer diagnosis and prognosis, and finally, treatment response and therapy resistance [42,43]. DNA methylation is one the most-commonly occurring epigenetic events, in which there is the addition of a methyl group to the carbon 5 position of cytosine within a cytosine guanine (CpG) dinucleotide by enzyme DNA methyltransferase. DNA methylation can be stable and heritable through cell divisions, but in the meantime, it is reversible and modifiable by specific enzymes [42]. Many studies report how breast cancer cells show disrupted methylation patterns in their DNA [39,44]. Moreover, the DNA methylation pattern can be very specific not only for different types of tumours (inter-tumour heterogeneity), but also for different tumour subgroups (intra-tumour heterogeneity) and, therefore, has been used also to identify different cancer types and to trace the primary origin of metastatic tumours [42,44].

In general, global DNA hypomethylation has been associated with cancer. DNA hypomethylation can determine chromosomal instabilities and gene activation, thus leading to the upregulation or overexpression of proto-oncogenes and increased recombination and mutation rates [44]. Hypomethylation contributes to oncogenesis also by the activation of latent retrotransposons or mobile DNA, such as long interspersed nuclear elements, which can determine the disruption of the expression of the adjacent gene, for example homeobox [42].

DNA hypermethylation in cancer, instead, is associated with a direct gene repression effect (of tumour-suppressor genes, for example), but also with compaction of chromatin, which in turn modifies its accessibility and, finally, determines the instability and alteration of gene expression (silencing of DNA repair genes, for example) [44]. However, the inhibition or activation of transcription by methylation is dependent on the analysed DNA gene segment (promoter, TSS, or gene-body).

DNA hypermethylation of promoters’ transcription start sites (TSS) or enhancers contributes to reducing gene expression or silencing by interfering with the binding of specific transcription factors to their recognition sites or by binding of transcriptional repressors specific for the methylated sequence [43]. Estecio and Issa [19] underlined that CpG island promoters are the most-straightforward compartment to evaluate when searching for aberrant DNA methylation in cancer, above all considering that these CpG islands usually are unmethylated in normal cells (except for imprinted and X-chromosome inactivated genes). Therefore, they speculated that these abnormally methylated gene promoters (along with other regions with regulatory function) will likely be revealed as important players in tumour biology. They reported examples of promoter hypermethylation of the CDKN2A and MLH1 genes.

On the contrary, hypermethylation at gene-bodies is associated with active transcription and gene expression, as a result of mRNA expression studies (as the case of homeobox) [42]. It has been suggested that the sliding of RNA polymerase over the gene-body attracts DNA methyltransferase enzymes and, therefore, that DNA methylation in a gene-body is a consequence of transcription, rather than an active agent promoting it. Others suggested that methylation marks embedded in coding sequences are associated with the timing of transcription initiation events [45]. Moreover, differences in CpG methylation between exon and intron regions raise the possibility that gene-body methylation participates in splicing regulation [19]. Finally, the biological meaning of gene-body methylation still remains unclear, and more studies are needed to address this issue.

Methylation markers in intergenic regions are thought to have little impact on genome activity [45]. In this study, according to previous studies, we found the most-important differences of the epimutation score between the two groups of ER-positive and ER-negative breast cancer precisely in the promoters of specific genes belonging to a few pathways.

### 4.2. Main Pathways Identified by Our Analysis

Pathway-centric analysis, as opposed to a gene-centric one, allows identifying recurrent altered signalling or function in cancer, based on alterations found in different genes belonging to the same pathway, but not altered at equal frequencies [17]. Moreover, evaluating the burden of epimutations per gene region (TSS200, TSS1500, promoter, gene-body, and first-exon) and then using these data for gene enrichment pathway analysis permit capturing the biological process involved by these variations avoiding treatment of individual occurrences of epigenetic markers such as nucleotide polymorphisms (i.e., as epialleles), since it was observed that the methylation state of any particular nucleotide in the promoter, for example, is usually irrelevant and could represent statistically significant alterations, but functionally uninformative differences [45].

ESR-mediated signalling was identified as the pathway whose genes are overcharged by a higher epimutation score due to hypomethylation of the TSS1500 gene region (corresponding at least in part to the promoter region) in ER-positive BC vs. ER-negative BC. In light of the fact that a hypomethylated promoter could permit gene expression (even if it is not a unique condition), we interpreted this result as coherent with a higher activation of this pathway in the BC group that expresses ER. In this sense, previous studies centred on the role of the epigenetic control of ER function confirmed our results. This is indirectly suggested by many studies that report a higher hypermethylation status of the ER promoter in the group of ER-negative BC and that ER gene hypermethylation is associated with lacking ER gene expression [46,47,48,49]. Moreover, other studies confirmed, for example, that inhibition of DNA methyltransferase (DNMT) in ER-negative BC cells induces re-expression of oestrogen receptor-alpha [50,51].

The pathways identified by a higher epimutation score due to hypermethylation of the TSS1500 gene region in ER-positive BC vs. ER-negative BC belong to the following main groups: the Notch pathway, the SUMOylation pathway, and two ubiquitination protease signalling pathways. Other studies confirmed that, generally, the hypermethylated loci in ER-negative tumours were clustered closer to the transcriptional start site compared to ER-positive tumours [52] and that there was a cumulative effect of a very large number of epigenetic perturbations to be correlated specifically and in cis with hundreds of additional transcriptional changes [53].

Interestingly, the SUMOylation pathway and ubiquitination protease signalling pathway belong to the same kind of protein post-translational modifications.

The data on the role of signalling in BC are few and contrasting. Todorović-Raković found that raised serum IFN-
γ
 levels are associated independently with favourable disease outcomes in hormonally dependent breast cancer [54]. On the other side, Yu and colleagues found that IFN-
γ
 induces tumour resistance to anti-PD-1 immunotherapy in BC [55], and experiments on BC cells demonstrated that IFN-
γ
 could upregulate the expression of PD-L1, promote cell migration and transmission, and facilitate the epithelial–mesenchymal transformation of breast cancer cells [56].

The SUMOylation and the Notch signalling pathways are the other two pathways whose genes emerged as characterised by a higher epimutation score due to hypermethylation in the TSS1500 gene region in the ER-positive vs. ER-negative BCs. Since we performed a direct comparison of the two BC groups, we hypothesised that the presence of a higher hypermethylation of the gene region that overlaps the gene promoters corresponds to a general reduced gene expression (as discussed before) and, consequently, to a reduced activity of these two pathways in the ER-positive BCs. Moreover, based on the direct comparison between the two groups of BCs, we speculated that the relative hypomethylation in the ER-negative BCs could justify the hypothesis of the presence of a state of hyperactivation of these two pathways in ER-negative BC. The presence of a significant activity of these two pathways in the ER-negative BC group is not lacking, as discussed thereafter [57].

### 4.3. On the Role of SUMOylation and Notch Pathways in ER-Negative BC and Their Correlation with Epithelial–Mesenchymal Transition and Breast Cancer Stem Cells

Many studies suggest the existence of complex and intricate relations among the biological process of epithelial to mesenchymal transition (EMT) and cancer stem cell (CSC) phenotype. EMT is characterised by the acquisition of phenotypic plasticity and the stem-cell-like properties of the tumour cells, including cytoskeleton adjustment, loss of cell polarity, and loss of cell adhesion. During EMT, cells lose their epithelial features and markers—such as the cobblestone shape and E-cadherin expression—to acquire a mesenchymal phenotype—assuming a spindle shape and mesenchymal markers, such as vimentin and fibronectin [38,58]. These mesenchymal attributes permit cancer cells to develop new capabilities, such as migration and invasiveness, pro-survival ability, stemness, immunosuppression, and chemoresistance [59]. These characteristics can lead to the formation of CSCs, the maintenance of aggressiveness, the initiation of metastasis, and tumour relapse [60].

CSCs were identified for the first time in 2003 in human breast tumours (BCSCs), and since then, a growing amount of evidence has supported their role in breast cancer initiation, intratumoural heterogeneity, progression, disease recurrence, metastasis, and resistance to therapy [61]. Actually, the origin of CSCs is not clear. In particular, the two main hypotheses are that they are cells already present in the tumour from its origin, but in a state of quiescence, or alternatively, that they originate at a secondary time through a process of de-differentiation (for example, through a process of partial/total EMT). Finding a set of markers to identify and target these partial/total EMT cells could lead to understanding the origin of CSCs and their deregulated pathways and could be a strategy for the development of therapeutics blocking cancer invasion and dissemination [59].

The EMT and CSCs have been correlated with alterations of the Notch and SUMOylation pathways in ER-negative BC in many studies [38,61,62,63,64,65].

Numerous studies found that Notch signalling activation and protein SUMOylation may promote breast cancer tumourigenesis and progression by accelerating cell cycle transition and proliferation and facilitating tumour cell EMT in breast epithelial cells in vivo and in vitro [38,61,62,63,64,65,66].

Notch1 knockdown in breast cancer cells suppressed the EMT process, tumour growth, migration, and invasion using in vitro and in vivo models. Jagged1-mediated Notch signalling activation was able to activate the EMT process and increase migration and invasion in breast cancer, mainly through upregulation of N1ICD. Notch1 signalling is able to reverse the epithelial cobblestone morphology of the cells to the spindle mesenchymal one, to induce the switching of epithelial markers such as E-cadherin by the upregulation of SNAIL, SIP1/ZEB2, and SLUG (which are direct transcriptional repressors of E-cadherin) and the acquisition of mesenchymal markers such as vimentin, N-cadherin, and fibronectin to reduce invasion and migration [61,63,64,65]. On the contrary, activation of Notch signalling can be suppressed by EMT-inhibiting microRNAs such as miR-34 and miR-200 [64]. The role of Notch signalling in EMT corresponds to its promotion of invasive and metastatic phenotypes. Activation of Notch signalling in non-invasive breast cancer cells promotes cell invasion and migration, while inhibition of Notch in invasive cells reduces their invasive and migratory capacity [61,63,64,65], and Notch signalling is correlated with metastasis in vivo [67].

In the same way, SUMOylation participates directly in the modifications of many transcription factors (TFs) and in the activation of various signalling involved in the control of EMT [38,58]. Several transcriptional factors’ activity—including ZEB1, SNAIL, and TWIST—which regulate mesenchymal cell marker expression, such as CDH1 (the E-cadherin gene) and promote EMT—is directly or indirectly influenced by the SUMOylation pathway. ZEB1, one of the main TFs involved in EMT, has been reported to be regulated by SUMOylation through different mechanisms. SUMOylation of ZEB1, as well as its homologue ZEB2, inhibits E-cadherin expression and induces EMT. Moreover, silencing of SENP1 (which has also the function of peptidase, which causes the hydrolysis of SUMO bonds) decreases the ZEB1 protein level, suggesting that deSUMOylation of ZEB has a role in activating the TF [58]. By regulating numerous oncoproteins, ZEB1 plays an important role in metastasis. In the ER-negative basal-like breast cancer (BLBC), a breast cancer subtype enriched with the expression of mesenchymal genes and reduced expression of epithelial ones including E-cadherin [68], downregulation of CDH1 is mediated by ZEB1, which recruits DNMT1 (a DNA methyl-transferase enzyme) to the CDH1 promoter to maintain the methylation status in the promoter. These results suggest that ZEB1 could act as a transcriptional repressor and an epigenetic modulator to induce EMT in breast cancer [69]. A recent study demonstrated that also ZNF451, a SUMO2/3-specific E3 ligase, is a positive regulator of EMT through the SUMOylation of TWIST2 at the K129 residue. SUMOylation stabilises TWIST2 by inhibiting its ubiquitination and degradation and, consequently, promotes EMT [58]. Two prominent mesenchymal transcription factors, SLUG and TWIST1, are upregulated in cells that present mesenchymal characteristics. The expression levels of SLUG AND TWIST1 are highest in ER-negative claudin-low tumours, and both genes identify letrozole-resistant disease. SLUG accumulation in basal-like tumours is also associated with BRCA1 mutations [70].

Moreover, a direct correlation between the aberrant Notch and SUMOylation pathways and the triple-negative phenotype of BC has been found in many studies.

Notch signalling has been seen hyperactivated in TNBC and in ER-positive BC with poor prognosis or with a higher risk of relapse (which have many features in common with ER-negative BC). It was suggested that this hyperactivation could have an important role in EMT induction and BCSCs’ proliferation in TNBC [39], while in ER-positive BC, this could induce hormone-therapy resistance [71]. Clinical analyses showed that JAG1, as well as Notch1, Notch3, and Notch4 are overexpressed at high levels in TNBC and correlated with the aggressive, metastatic, and therapy-resistance phenotype characteristic of TNBC and are associated with poor clinical prognosis. Moreover, the expression of the Notch target, HES4, was correlated with poor prognosis outcomes in TNBC patients [63]. Reedijk and colleague [71] observed that patients with tumours expressing high levels of JAG1 or Notch1 had a significantly poorer overall survival compared with patients expressing low levels of these genes, and moreover, a synergistic effect of high-level JAG1 and high-level Notch1 coexpression on overall survival was observed. Therefore, they suggested a mechanism whereby Notch is activated in aggressive and poor-prognosis breast tumours (since JAG1 is a ligand of Notch-receptor-1) and that the basal breast cancer subgroup (belonging to ER-negative BC) shows poor overall survival as a result of JAG1-induced Notch activation in some of these tumours. Reference [72] performed exome sequencing analysis to identify Notch mutations in various solid tumours, revealing that constitutive receptor activation induced by Notch1 and Notch2 mutations is limited to TNBC. A TNBC cell line with Notch1 rearrangement also exhibited high-level Notch1 intracellular domain (N1ICD) accumulation with subsequent upregulated target gene expression. In addition, Notch1 or Notch2 mutations can synergistically act with EZH2 to inhibit the tumour suppressor PTEN’s transcription at the promoter in TNBC [73].

In a gene expression study, Orzechowska M. and colleagues evaluated [37] the effect of the differential expression of Notch members on DF in luminal type A (lumA) and triple-negative (TN) BC. This study highlighted significant differences in the biology of the two tumours and indicated differences in the signals activating the Notch pathway and in particular suggested a role of Notch signalling in BRCA progression through triggering EMT. From their analysis, it emerged that aberrant expression and regulation of Notch receptors have the most-significant influence on the course of the disease. Notably, their results indicated that, while there are subgroups of patients who will probably never experience disease relapse, other subgroups exist within the ER-positive lumA subtype that have a higher risk of recurrence due to potential transition into the mesenchymal cell type. Moreover, their findings indicate that the expression profiles of Notch pathway members can be used to differentiate the DFS in lumA and TNBC subtypes and, so, may serve as novel prognostic biomarkers. Finally, they highlighted that MMP11, TAGLN, and THB2, three genes involved in acquiring the mesenchymal phenotype and that are regulated by the Notch pathway, can be used as potential therapeutic targets.

On the other hand, also the SUMOylation pathway seems to be involved in the maintenance of the characteristics of TNBC and the basal BC subtype (belonging to the ER-negative BC group). Bogacheck and colleagues demonstrated that inhibition of the SUMOylation pathway reduced cell invasiveness and induced functional loss of CSCs in basal BC [74]. Moreover, the same group in another study [62] established that SUMOylation inhibitors induce a basal-to-luminal transition in BC cells and inhibit tumour outgrowth of basal cancer xenografts. Wang Q and colleagues reached similar conclusions about the relation of SUMOylation and ER-negative BC, evaluating the role of SUMO1-activating enzyme subunit1 (SAE1), an E1-ligase-activating enzyme, indispensable for protein SUMOylation in TNBC. They found that mRNA and protein SAE1 expression is increased in TNBC tissues compared to adjacent normal tissue and their expression levels are significantly associated with overall survival (OS) and disease-free survival (DFS) [75]. In the review by Zhu et al., the multiple ways through which the SUMOylation pathway can influence stem cell functions in cancer were recapitulated [76].

Finally, we discuss the role of epigenetic control on the Notch and SUMOylation pathways. Interestingly, DNA methylation has been confirmed to have an important role in the regulation of the Notch and SUMOylation pathways. Yousefi and colleagues, using the TCGA HumanMethylation450 Array data, determined that the epigenetic regulation of the Notch regulators contributes to their expression and suggested that Notch receptors and ligands’ expression is generally associated with the tumour subtype, grade, and stage [77]. Aithal et al. focused on the methylation status of genes in the Notch signalling pathway from various cancers and highlighted how this epigenetic alteration can be used as a biomarker for cancer diagnosis and subsequent treatment [78]. Accordingly, due to the important role of epigenetic reprogramming and DNA methylation, Hanif and colleagues highlighted how these processes could be determinant specifically in TNBC, in which we have seen that the Notch pathway could have fundamental regulatory functions [39]. Finally, Kagara et al. demonstrated that methylation is a significant mechanism regulating CD44, CD133, and Musashi-1, which are specific BCSC-related genes, and that the hypomethylation of these genes correlates with a significant inverse correlation of mRNA expression in the TNBC subtype [79].

We want also to discuss the limits of our study. First and foremost, for one dataset (GSE69914), the patients’ age and tumour stage were not available; therefore, age was inferred through methylation data, while tumour stage was omitted in the analysis of that dataset. Second, we introduced an epimutation score based on quantile ranking of the difference in the methylation levels; this is a new method of analysis that needs to be validated with other studies. Finally, in the discussion, we interpreted the results of the hypomethylation of the genes of ESR-mediated signalling in ER-positive BC as corresponding to a higher expression of the genes in this group of BC. Yet, we know that this condition of hypomethylation is not sufficient to draw this conclusion. An analogous consideration could be drawn when we considered the hypermethylation promoter of genes belonging to the Notch and SUMOylation pathways in ER-positive BC. In this case, we concluded that the hypermethylation in ER-positive BC corresponds to a reduced methylation in ER-negative BC (since we performed a direct comparison of methylation data between these two groups of BCs); we considered this condition potentially correlated with a higher expression of these genes in this group of ER-negative BCs. We know that these are only indirect hypotheses that need to be confirmed.

## Figures and Tables

**Figure 1 cancers-15-04109-f001:**
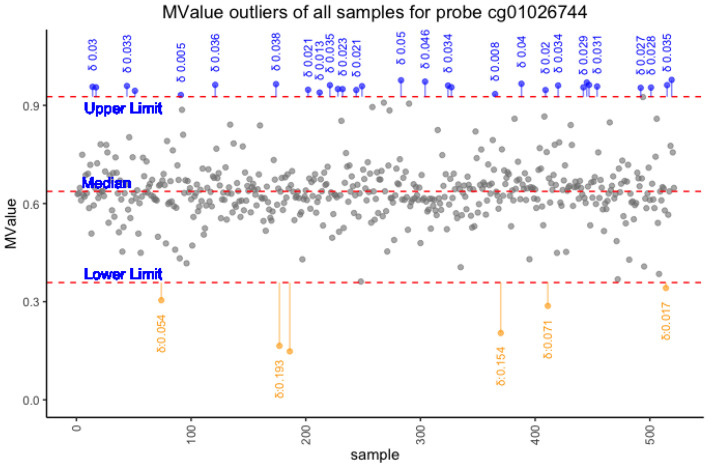
The figure shows the ranges used to define the epimutations; the upper part of the figure shows the epimutations due to hypermethylation, and the lower part shows two epimutations due to hypomethylation.

**Figure 2 cancers-15-04109-f002:**
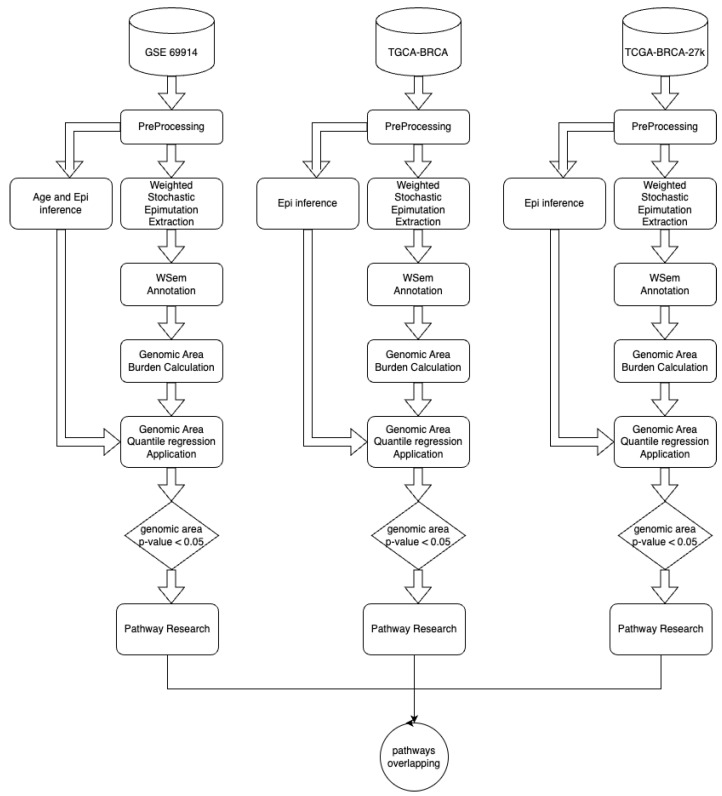
Pipelines adopted to conduct the study.

**Figure 3 cancers-15-04109-f003:**
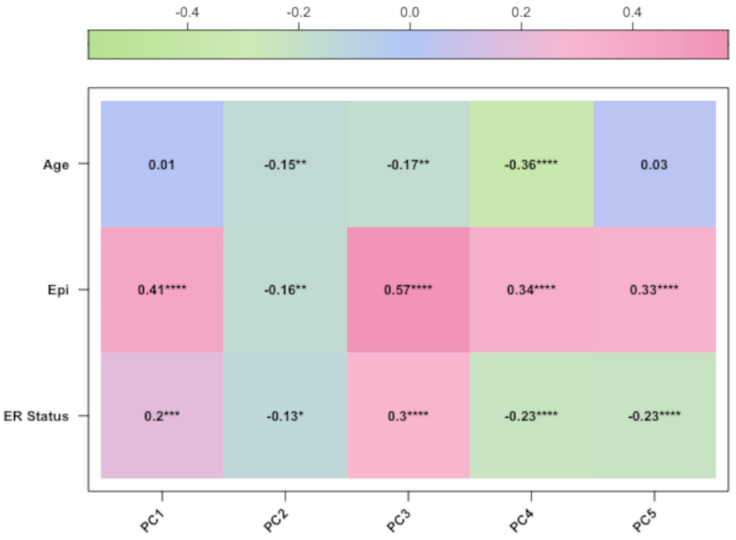
The corplot diagram of the correlation test between each principal component and phenotypic trait for the GSE69914 study. Asterisks have the following correspondence *p*-value ’****’ = 0, ’***’ = 0.0001, ’**’ = 0.001, ’*’ = 0.01, ‘ ’ = 0.05.

**Figure 4 cancers-15-04109-f004:**
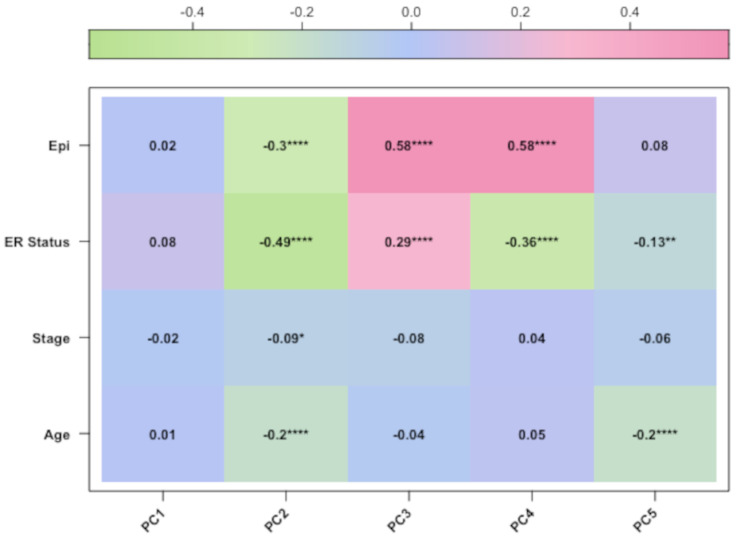
The corplot diagram of the correlation test between each principal component and phenotypic trait for the TCGA-BRCA study. Asterisks have the following correspondence *p*-value ’****’ = 0, ’**’ = 0.001, ’*’ = 0.01, ‘ ’ = 0.05.

**Figure 5 cancers-15-04109-f005:**
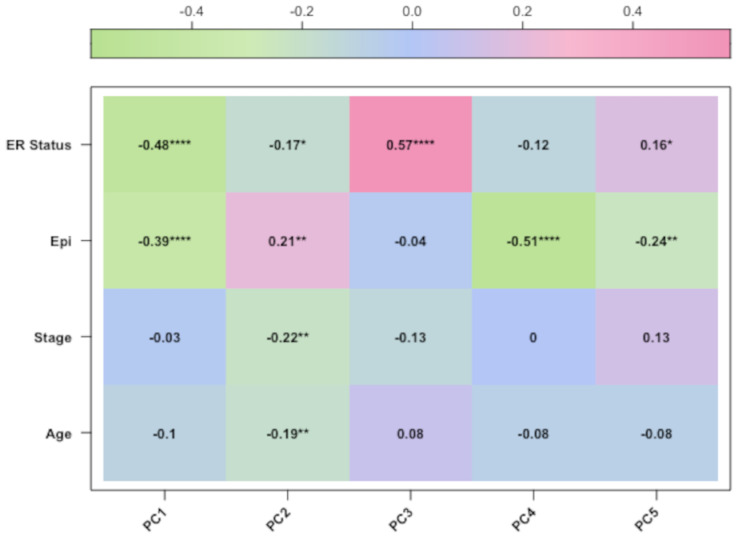
The corplot diagram of the correlation test between each principal component and phenotypic trait for the TCGA-BRCA-27k study. Asterisks have the following correspondence *p*-value ’****’ = 0, ’**’ = 0.001, ’*’ = 0.01, ‘ ’ = 0.05.

**Figure 6 cancers-15-04109-f006:**
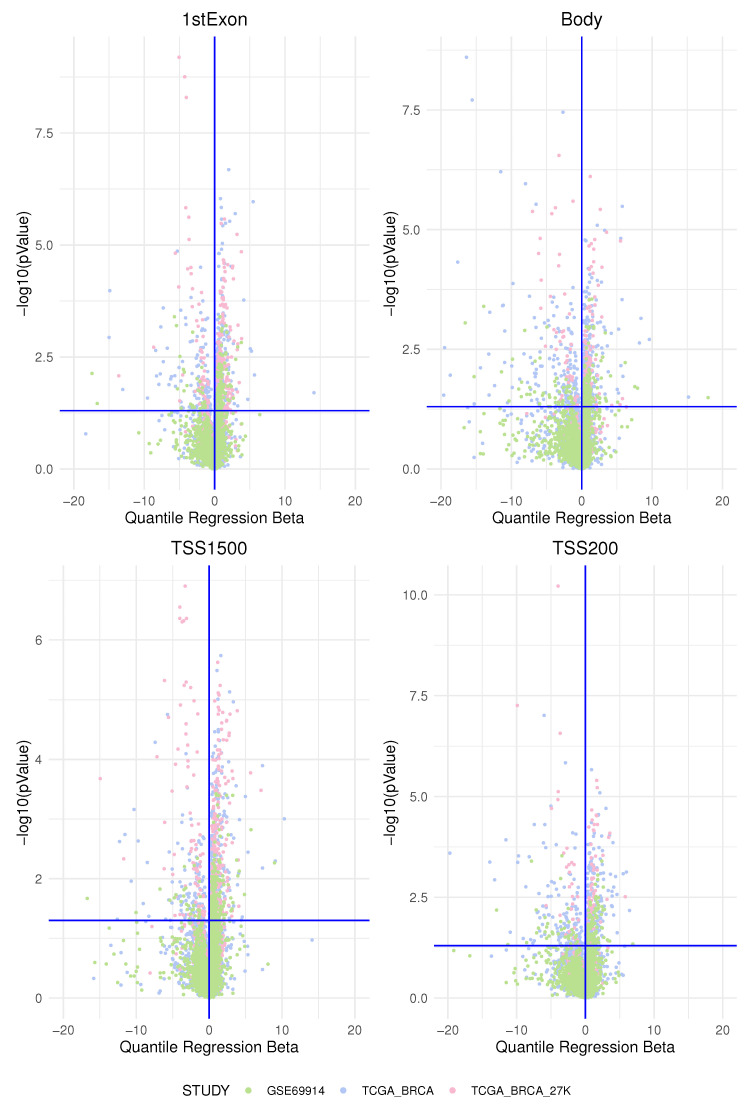
The figure represents the volcano plot for each genomics area: each dot represents, for each gene, the quantile regression beta coefficient and the corresponding 
−log10(pValue)
. All four plots represent the genes affected by an epimutation score due to hypermethylation. The dot’s colors are associated to a studied data-set as in the legend.

**Figure 7 cancers-15-04109-f007:**
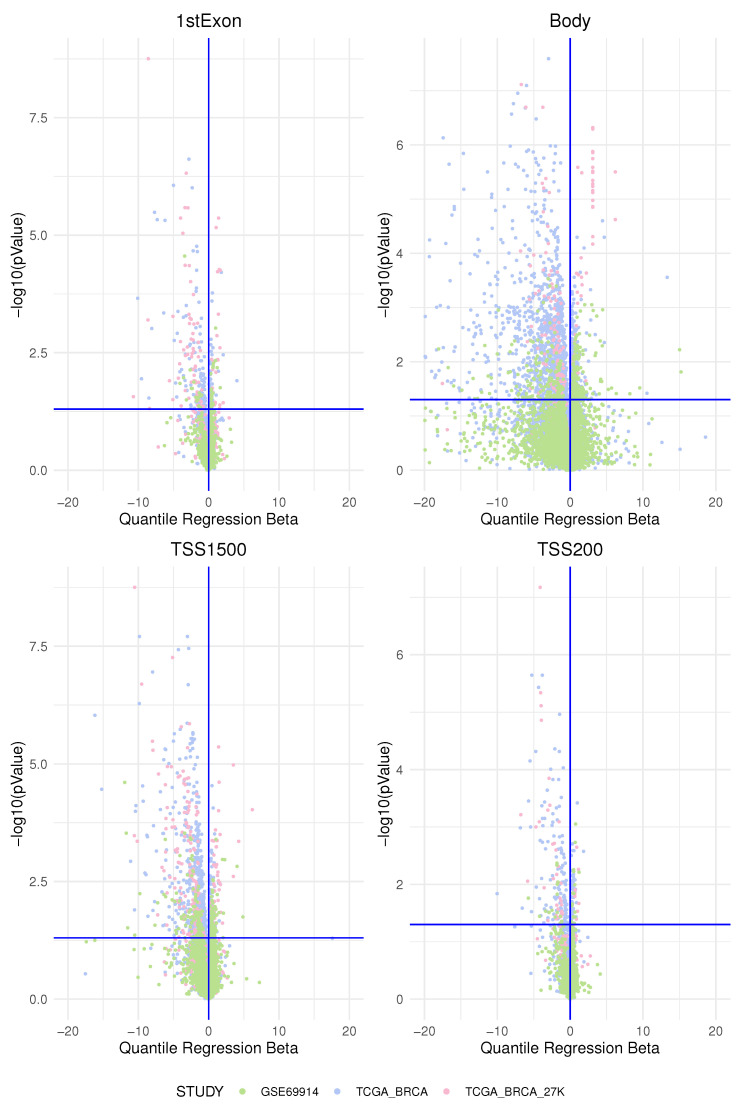
The figure represents the volcano plot for each genomics area: each dot represents, for each gene, the quantile regression beta coefficient and the corresponding 
−log10(pValue)
. All four plots represent the genes affected by an epimutation score due to hypomethylation. The dot’s colors are associated to a studied data-set as in the legend.

**Figure 8 cancers-15-04109-f008:**
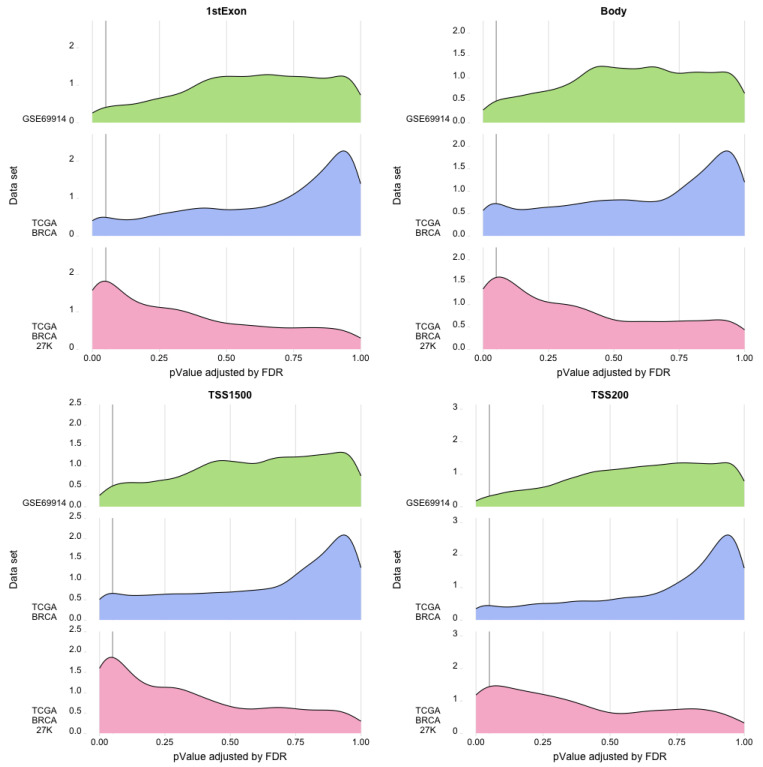
Ridge plots. Each plot represents for each dataset the density (*y*-axis) of the *p*-value (*x*-axis) associated with the gene area with an epimutation burden due to hypermethylation.

**Figure 9 cancers-15-04109-f009:**
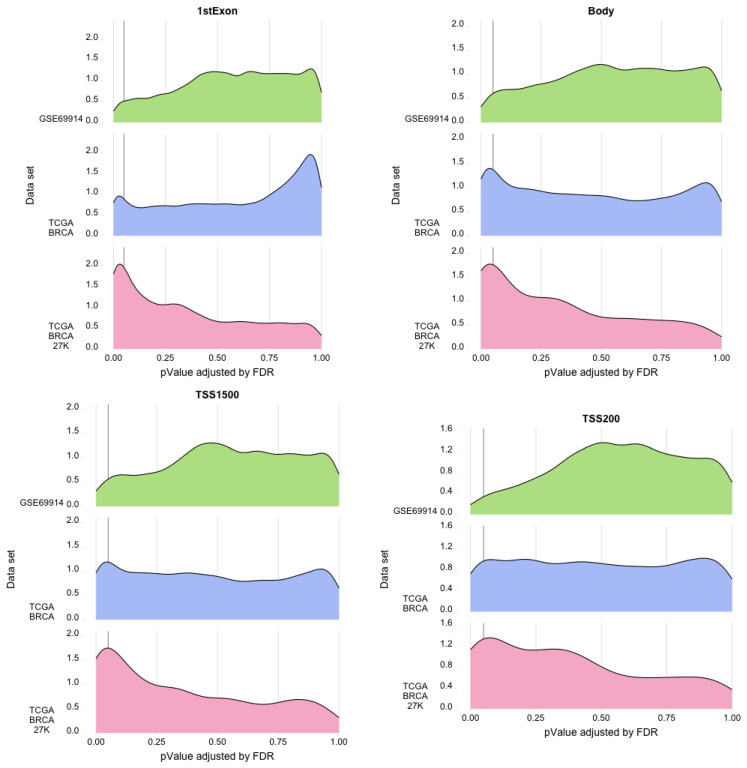
Ridge plots. Each plot represents for each dataset the density (*y*-axis) of the *p*-value (*x*-axis) associated with the gene area with an epimutation burden due to hypomethylation.

**Figure 10 cancers-15-04109-f010:**
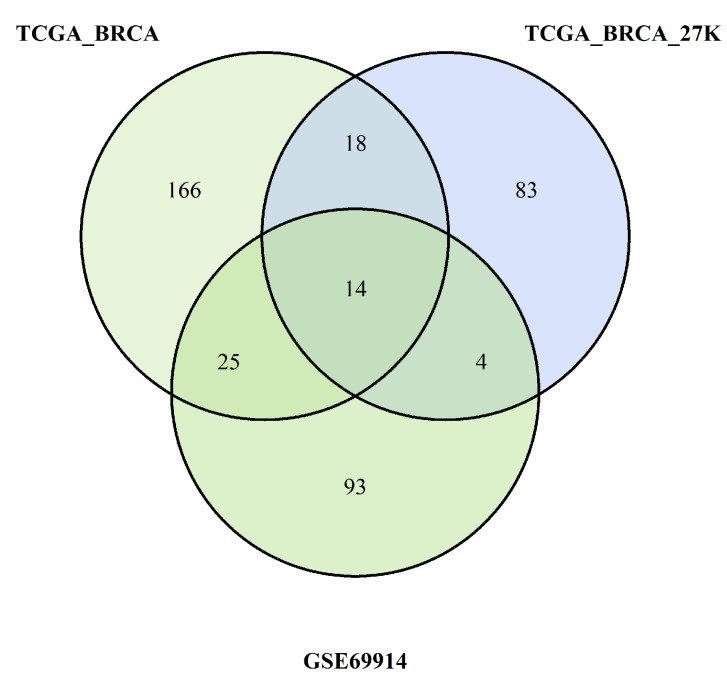
Pathways overlapping among the three studies due to burden of probes with hypermethylation.

**Figure 11 cancers-15-04109-f011:**
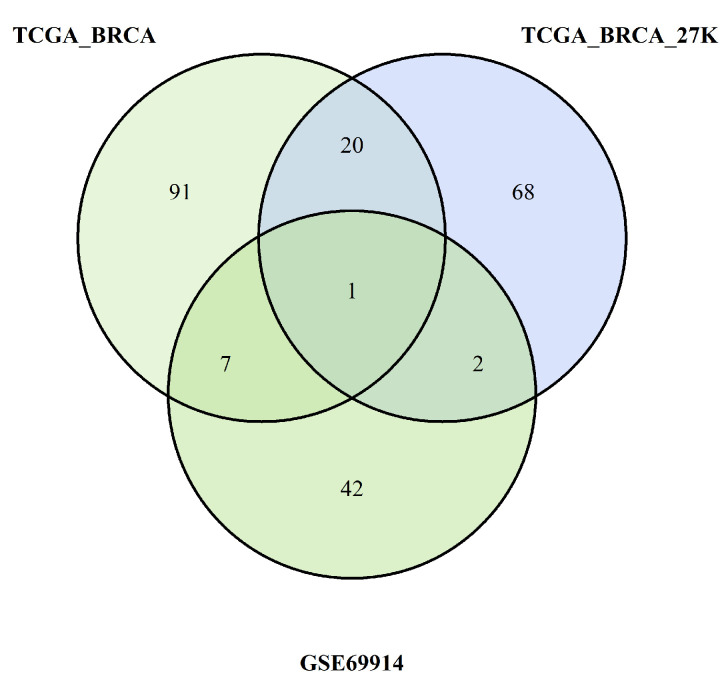
Pathways overlapping among the three studies due to burden of probes with hypomethylation.

**Table 1 cancers-15-04109-t001:** Characteristics of the three datasets used for our analysis.

Dataset Name	Number of Samples	Age Available	Technology	Format	Number of Probes (% of Total)
TCGA-BRCA [20]	521	Yes	Human Methylation 450	Raw (idat files)	385,578 (79.41%)
TCGA-BRCA-27K [20]	180	Yes	Human Methylation 27	Raw (idat files)	25,522 (92.54%)
GSE69914 [21]	302	No	Human Methylation 450	TXT (beta-value matrix)	290,250 (59.78%)

**Table 2 cancers-15-04109-t002:** The clinical characteristics of the patients and the tumour samples of the three datasets used for our analysis.

Dataset	Age (std)	Stage I	Stage II	Stage III	Stage IV	ER+	ER−	*n* (Patients)
TCGA-BRCA	57.65 (12.74)	91 (17%)	281 (54%)	145 (28%)	4 (0.77%)	426 (82%)	95 (18%)	521 (52%)
GSE69914	49.93 (5.62)	-	-	-	-	254 (84%)	48 (16%)	302 (30%)
TCGA-BRCA-27k	59.19 (13.11)	46 (26%)	107 (59%)	20 (11%)	7 (3.89%)	140 (78%)	40 (22%)	180 (18%)

**Table 3 cancers-15-04109-t003:** HER2 distribution in the breast cancers of the 450k-TCGA dataset.

	ER−	ER+	Total
Undetermined	31 (32.5%)	128 (30%)	159
HER2−	55 (58%)	244 (57.3%)	299
HER2+	9 (9.5%)	54 (12.7%)	63
Total	95 (100%)	426 (100%)	521

**Table 4 cancers-15-04109-t004:** Pathways impacted by SEM caused by hypomethilation for dataset GSE69914 27k on the TSS1500 gene area.

Description	Regression Beta	Genes with Increased Burden	Genes with Decreased Burden
Integrin signalling	5.06		BCAR1

**Table 5 cancers-15-04109-t005:** Pathways impacted by SEM caused by hypomethilation for dataset TCGA-BRCA 27k on the TSS1500 gene area.

Description	Regression Beta	Genes with Increased Burden	Genes with Decreased Burden
Integrin signalling	7.12	FGB	SRC

**Table 6 cancers-15-04109-t006:** Pathways impacted by SEM caused by hypomethilation for dataset TCGA-BRCA on the TSS1500 gene area.

Description	Regression Beta	Genes with Increased Burden	Genes with Decreased Burden
Integrin signalling	3.47	AKT1	RAP1A, SRC

**Table 7 cancers-15-04109-t007:** Pathways impacted by SEM caused by hypermethilation for dataset GSE69914 on the TSS1500 gene area.

Description	Regression Beta	Genes with Increased Burden	Genes with Decreased Burden
SUMOylation of Transcription Factors	6.33	PIAS1	
RHOJ GTPase Cycle	4.60	CAV1, DEPDC1B	
Constitutive Signalling by Notch1 HD + PEST Domain Mutants	4.36	HEY2, PSEN1	
Constitutive Signalling by Notch1 PEST Domain Mutants	4.36	HEY2, PSEN1	
Signalling by Notch1 HD + PEST Domain Mutants in Cancer	4.36	HEY2, PSEN1	
Signalling by Notch1 PEST Domain Mutants in Cancer	4.36	HEY2, PSEN1	
Signalling by Notch1 in Cancer	4.36	HEY2, PSEN1	
RHOQ GTPase Cycle	4.29	CAV1, DEPDC1B	
RHOG GTPase Cycle	3.52	CAV1, DEPDC1B	
Signalling by Notch1	3.42	HEY2, PSEN1	
RAC2 GTPase Cycle	2.94	CAV1, DEPDC1B	
RAC3 GTPase Cycle	2.75	CAV1, DEPDC1B	
SUMO E3 Ligases SUMOylate Target Proteins	2.32	DDX17, PIAS1	RARA
SUMOylation	2.23	DDX17, PIAS1	RARA

**Table 8 cancers-15-04109-t008:** Pathways impacted by SEM caused by hypermethilation for dataset TCGA-BRCA 27k on the TSS1500 gene area.

Description	Regression Beta	Genes with Increased Burden	Genes with Decreased Burden
SUMOylation of Transcription Factors	5.93	TFAP2C	TFAP2B
RHOQ GTPase Cycle	5.02	CDC42BPA, CDC42EP3, OBSCN, SYDE1	PREX1
Constitutive Signalling by Notch1 HD + PEST Domain Mutants	4.09	HDAC1, HDAC3, HDAC9, JAG2	
Constitutive Signalling by Notch1 PEST Domain Mutants	4.09	HDAC1, HDAC3, HDAC9, JAG2	
Signalling by Notch1 HD + PEST Domain Mutants in Cancer	4.09	HDAC1, HDAC3, HDAC9, JAG2	
Signalling by Notch1 PEST Domain Mutants in Cancer	4.09	HDAC1, HDAC3, HDAC9, JAG2	
Signalling by Notch1 in Cancer	4.09	HDAC1, HDAC3, HDAC9, JAG2	
RHOG GTPase Cycle	3.29	ARHGDIG, DOCK3, EPHA2	PREX1
RHOJ GTPase Cycle	3.23	CDC42BPA, SYDE1	PREX1
Signalling by Notch1	3.20	HDAC1, HDAC3, HDAC9, JAG2	
RAC2 GTPase Cycle	2.76	DOCK3, EPHA2, SYDE1	PREX1
RAC3 GTPase Cycle	2.58	EPHA2, NOX1, SYDE1	PREX1
SUMO E3 Ligases SUMOylate Target Proteins	2.17	CTBP1, HDAC1, L3MBTL2, TFAP2C	DNMT3B, TFAP2B
SUMOylation	2.09	CTBP1, HDAC1, L3MBTL2, TFAP2C	DNMT3B, TFAP2B

**Table 9 cancers-15-04109-t009:** Pathways impacted by SEM caused by hypermethilation for dataset TCGA-BRCA on the TSS1500 gene area.

Description	Regression Beta	Genes with Increased Burden	Genes with Decreased Burden
SUMOylation of Transcription Factors	3.43	CDKN2A, PIAS1	
RHOJ GTPase Cycle	3.11	FMNL3, PAK1, PAK2, RHOJ	CDC42BPB
RHOQ GTPase Cycle	2.90	ARHGAP17, ARHGAP33, PAK1, PAK2	CDC42BPB
RAC2 GTPase Cycle	2.79	ARHGAP17, BAIAP2L1, DOCK3, LBR, PAK1, PAK2, VRK2	
Constitutive Signalling by Notch1 HD + PEST Domain Mutants	2.36	DLL1, HDAC1, HDAC4, PSEN1	
Constitutive Signalling by Notch1 PEST Domain Mutants	2.36	DLL1, HDAC1, HDAC4, PSEN1	
Signalling by Notch1 HD + PEST Domain Mutants in Cancer	2.36	DLL1, HDAC1, HDAC4, PSEN1	
Signalling by Notch1 PEST Domain Mutants in Cancer	2.36	DLL1, HDAC1, HDAC4, PSEN1	
Signalling by Notch1 in Cancer	2.36	DLL1, HDAC1, HDAC4, PSEN1	
RAC3 GTPase Cycle	2.23	ARHGAP17, BAIAP2L1, LBR, PAK1, PAK2, VRK2	
RHOG GTPase Cycle	1.90	DOCK3, LBR, PAK2, VRK2	
Signalling by Notch1	1.85	DLL1, HDAC1, HDAC4, PSEN1	
SUMO E3 Ligases SUMOylate Target Proteins	1.67	CBX2, CDKN2A, DDX17, HDAC1, HDAC4, NR3C1, PIAS1	RARA
SUMOylation	1.61	CBX2, CDKN2A, DDX17, HDAC1, HDAC4, NR3C1, PIAS1	RARA

**Table 10 cancers-15-04109-t010:** Pathways shared for the TSS1500 gene area due to hypermethylation.

REACTOME-ID	Description
R-HSA-2894862	Constitutive Signalling by Notch1 HD + PEST Domain Mutants
R-HSA-2644606	Constitutive Signalling by Notch1 PEST Domain Mutants
R-HSA-2894858	Signalling by Notch1 HD + PEST Domain Mutants in Cancer
R-HSA-2644602	Signalling by Notch1 PEST Domain Mutants in Cancer
R-HSA-2644603	Signalling by Notch1 in Cancer
R-HSA-1980143	Signalling by Notch1
R-HSA-157118	Signalling by Notch
R-HSA-3232118	SUMOylation of Transcription Factors
R-HSA-4551638	SUMOylation of Chromatin Organisation Proteins
R-HSA-3108232	SUMO E3 Ligases SUMOylate Target Proteins
R-HSA-2990846	SUMOylation
R-HSA-877312	Regulation of IFNG Signalling
R-HSA-5689603	UCH Proteinases
R-HSA-5689880	Ub-Specific Processing Proteases

**Table 11 cancers-15-04109-t011:** Pathways shared for the TSS1500 gene area due to hypomethylation.

REACTOME-ID	Description
R-HSA-8939211	ESR-Mediated Signalling

## Data Availability

Data are available from the respective web sites for GSE69914 [21] and TGCA-BRCA and TCGA-BRCA-27k [20].

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
