# Peer review of "Notch, SUMOylation, and ESR-Mediated Signalling Are the Main Molecular Pathways Showing Significantly Different Epimutation Scores between Expressing or Not Oestrogen Receptor Breast Cancer in Three Public EWAS Datasets"

_cancers, 2023, doi:10.3390/cancers15164109_

Round 1

Reviewer 1 Report

The authors explored the role of the altered methylation profile in regulating signalling pathways with specific functions in the two subgroups of ER-BC.This article has some research value, but it is recommended to adjust the logical structure of the article to better support its conclusions.In addition, the format of references needs to be standardized, such as omitting some authors if the number of authors exceeds a certain limit. I don't have any more comments.

None.

Author Response

Response to Reviewer 1 Comments

Point 1: The authors explored the role of the altered methylation profile in regulating signalling pathways with specific functions in the two subgroups of ER-BC. This article has some research value, but it is recommended to adjust the logical structure of the article to better support its conclusions. In addition, the format of references needs to be standardized, such as omitting some authors if the number of authors exceeds a certain limit. I don't have any more comments.

Response 1:

Thanks to the reviewer we improved the description of our workflow. We charted the logical structure of the study  using a workflow graph in order to make it more understandable.

We defined a better definition of epigenetic variations so that it is simpler to understand the definition of epigenetic score. In the meanwhile, the added flowchart illustrates the next logical steps through which the methylation data were processed.

We standardized the format of references, leaving the first author followed by et. al. as can be seen in the new bibliography pages 21-23.

Reviewer 2 Report

Corsaro and Gambino present an analysis of three breast cancer datasets looking for methylation differences between ER+ and ER- cancers. They claim to identify enriched hypomethylation of estrogen receptor signaling pathway promoters in ER+ tumors, and hypermethylation of Notch pathway and SUMOylation-related gene promoters in ER+ tumors.

     However, there are multiple issues with this paper in its current form that need to be addressed.

     1. Clarity. For example, it is unclear whether the term 'epimutation' as used by the authors means a change in the average methylation of a specific nucleotide, or of a gene region, or whether this term had a different specific meaning. Lines 47-52 in the manuscript seem to distinguish between differential methylation versus 'epimutation', without making clear how they are different. This is a key term used in this manuscript, so it was extremely frustrating to not have a clear definition of how the authors meant it to be defined. Their description of their epimutation analysis method (lines 147-169) was also difficult to follow; it seems the top 20% highest and lowest methylation frequencies for each gene site were called as epimutations, but I may be very mistaken in this understanding due to the unclear description. Figures 6 and 7 are similarly opaque - what does the Y-axis indicate in these figures?

2. Baseline. If I understand what the authors did (see point 1, above), they compared the frequency of methylation of given genes' regions (TSS200, gene body, etc.) between ER+ versus ER- tumors. Was there any consideration given to normal tissue gene methylation levels? If yes, this needs to be made clear; if no, then the changes observed cannot be properly interpreted - they are only relative changes between two groups of tumors, with no relationship to normal tissue. (For example, a hypothetical gene X may be methylated in 10% of normal tissue cells, 60% or ER+ tumor cells, and 90% of ER- tumor cells. But simply comparing ER+ to ER- would give a 50% change in methylation frequency, completely missing the six-fold difference between ER+ and normal baseline.)

3. Validation. For any biological significance to be derived from this manuscript, the authors need to demonstrate that the identified methylation changes correspond to different protein expression or pathway activity levels.

4. Sub-typing. Breast cancer has several recognized, well-established sub-types: Luminal A, Luminal B, Basal, Her2+. These sub-types have distinct prognostic attributes, and do not directly correspond to the broader ER+/ER- characterization (especially the Her2+ category spans the ER+/- axis). The authors need to refine their analysis, at minimum separating out Her2+ tumors into a category of their own to reflect the realities of clinical tumor behavior.

5. Age characterization. The authors note that age data was not available for one dataset, and describe a method to determine age from methylation data. However, they used only cancer samples in this determination. Oncogenesis itself will disrupt normal cellular methylation patterns, calling into question the appropriateness of this technique. Furthermore, the authors then evaluate the correlation between methylation profile and patient age (line 204-5) - WHICH WAS DETERMINED BASED ON THE METHYLATION PROFILE! This seems doubly problematic - of course they are going to be correlated! For studies on methylation distribution, a method of determining sample age independent of methylation status is needed.

The paper could use proof-reading, as I identified many minor errors in language (lines 35, 47, 62, 78 on page 2 alone); however, the English was generally comprehensible.

Author Response

Response to Reviewer 2 Comments

 We thank the reviewer for her/his feedbacks.

Point 1: Clarity. For example, it is unclear whether the term 'epimutation' as  used by the authors means a change in the average methylation of a specific nucleotide, or of a gene region, or whether this term had a different specific meaning. Lines 47-52 in the manuscript seem to distinguish between differential methylation versus 'epimutation', without making clear how they are different. This is a key term used in this manuscript, so it was extremely frustrating to not have a clear definition of how the authors meant it to be defined. Their description of their epimutation analysis method (lines 147-169) was also difficult to follow; it seems the top 20% highest and lowest methylation frequencies for each gene site were called as epimutations, but I may be very mistaken in this understanding due to the unclear description. 

Response 1: A SEM (stochastic epigenetic mutation or “epimutation”), at a given CpG site, was defined as an extreme outlier of DNA methylation value distribution across individuals [Gagliardi et al., 2020]. Therefore, differential methylation comparison, based on different average methylation values, is a different concept. We add the definition of SEM in the introduction of our article to clarify the concept.

The paper was updated at paragraph 2.3 to clarify how the epimutation’s definition, moreover a new plot to represent how the epimutation are calculated was added.

Point 2: Figures 6 and 7 are similarly opaque - what does the Y-axis indicate in these figures?

Response 2: The plots are produced with a y-axis explaining the density of the p-value, The caption was rewritten to clarify better what the plots represent.

Point 3: Baseline. If I understand what the authors did (see point 1, above), they compared the frequency of methylation of given genes' regions (TSS200, gene body, etc.) between ER+ versus ER- tumors. Was there any consideration given to normal tissue gene methylation levels? If yes, this needs to be made clear; if no, then the changes observed cannot be properly interpreted - they are only relative changes between two groups of tumors, with no relationship to normal tissue. (For example, a hypothetical gene X may be methylated in 10% of normal tissue cells, 60% or ER+ tumor cells, and 90% of ER- tumor cells. But simply comparing ER+ to ER- would give a 50% change in methylation frequency, completely missing the six-fold difference between ER+ and normal baseline.)

Response 3: Thanks to the observation of the reviewer, that allows us to better focus the aim of our study. We excluded the comparison with normal tissue that is already present in other studies. The aim of our study is to elucidate if the presence of SEM has a role on some molecular pathway between the two subgroups of BC identified by the expression of ER. This type of comparison between these two groups of BC, neglecting normal tissue was already performed (for differential mean methylation values or differential mRNA expression), but not for the presence of SEM [see article by Orzechowska et al.].

Point 4: Validation. For any biological significance to be derived from this manuscript, the authors need to demonstrate that the identified methylation changes correspond to different protein expression or pathway activity levels.

Response 3: Thanks to the observation of the reviewer. We cited this observation at the end of our study as a limit because we have not yet demonstrated change in the expression of mRNA or protein. But this is a characteristic of other studies (Gentilini et al., Teschendorff et al.) that focalize their attention on methylation aspects and we did the same. We underline in our conclusion that this could be a next step to be explored.

Point 5:  Sub-typing. Breast cancer has several recognized, well-established sub-types: Luminal A, Luminal B, Basal, Her2+. These sub-types have distinct prognostic attributes, and do not directly correspond to the broader ER+/ER- characterization (especially the Her2+ category spans the ER+/- axis). The authors need to refine their analysis, at minimum separating out Her2+ tumors into a category of their own to reflect the realities of clinical tumor behavior.

Response 5: In the introduction we used this phrase to take in consideration this aspect: “Although the two BC groups identified in this way (ER positive/ ER 35 negative) are not completely homogeneous, they differ in their biological characteristics, clinical behaviour and therapeutic options”. The choice of using only the expression of ER to identify the two subgroups was taken to simplify the experiment but this not compromise the results since her2+ BC are only a minimal part in both groups (HER2+ BC are present in low number in both the groups ER+ and ER-) and this does not influence our analysis. In fact other studies that report direct comparison of BC with or without ER, often neglect HER2 expression. We added a comment (row 209) and a table (table3) where we analyze the ER and HER2 relationship to confirm what we supposed.

Point 6:  Age characterization. The authors note that age data was not available for one dataset, and describe a method to determine age from methylation data. However, they used only cancer samples in this determination. Oncogenesis itself will disrupt normal cellular methylation patterns, calling into question the appropriateness of this technique. Furthermore, the authors then evaluate the correlation between methylation profile and patient age (line 204-5) - WHICH WAS DETERMINED BASED ON THE METHYLATION PROFILE! This seems doubly problematic - of course they are going to be correlated! For studies on methylation distribution, a method of determining sample age independent of methylation status is needed.

 Response 6: Thanks to the reviewer. We added to the article that for the third dataset (the GSE69914), the PCA correlation between age and methylation profile was shown only for completeness, since age was calculated on methylation data. However, we decided to maintain the age in our regression model since the age was determined using a convolution model with a subset of around 30 probes over more than at least 27000 probes present in all the dataset after the quality control. 

Point 7 English: The paper could use proof-reading, as I identified many minor errors in language (lines 35, 47, 62, 78 on page 2 alone); however, the English was generally comprehensible.

Response 7: Thanks to the reviewer. We tried to correct all the mistakes we noted. For other mistakes we did not correct, probably we have written in English UK and not in American English?